

# Effects of dry and wet Saharan dust deposition in the tropical North Atlantic Ocean

Laura F. Korte[1], Franziska Pausch[2], Scarlett Trimborn[2], Corina P. D. Brussaard[3], Geert-Jan A. Brummer[1], Michèlle van der Does[1], Catarina V. Guerreiro[4], Laura T. Schreuder[3], Chris I. Munday[1], Jan-Berend W. Stuut[1,5]

[1]NIOZ Royal Netherlands Institute for Sea Research, Department of Ocean Systems, and Utrecht University, Texel, the Netherlands

[2]Alfred Wegener Institute for Polar and Marine Research, Am Handelshafen 12, 27570 Bremerhaven, Germany

[3]NIOZ Royal Netherlands Institute for Sea Research, Department of Marine Microbiology and Biogeochemistry, and Utrecht University, Texel, the Netherlands

[4]MARE, Marine and Environmental Science Centre, Faculdade de Ciências da Universidade de Lisboa, Campo Grande, 1749-016 Lisboa, Portugal

[5]MARUM Center for Marine Environmental Sciences, University of Bremen, Bremen, Germany

*Correspondence to*: Laura F. Korte (laura.korte@nioz.nl)

**Abstract.** Incubation experiments comprising Saharan dust additions were conducted in the tropical North Atlantic Ocean along an east-west transect at 12°N to study the phytoplankton response to nutrient release in oligotrophic seawater conditions. Experiments were performed at three stations (M1, M3, M4), mimicking wet and dry deposition of low and high amounts of Saharan dust deposition from two different dust sources (paleo-lake and sand dune). Dust particle sizes were adjusted to resemble dust that is naturally deposited over the ocean at the experiment sites. For wet dust deposition, the dust was pre-leached in acidified 'artificial rainwater' ($H_2SO_4$) for 16 to 24 hours, mimicking acid cloud processing at different pH values. Experiments were run up to eight days. Daily nutrient measurements of phosphate ($PO_4^{3-}$), silicate ($SiO_4^{4-}$), nitrate ($NO_3^-$) and cell abundances were performed in addition to measurements of concentrations of total dissolved iron (DFe), particulate organic carbon (POC), and dissolved inorganic carbon (DIC) at the start and at the end of the experiments.

A significant initial increase and subsequent gradual decrease in $PO_4^{3-}$, $SiO_4^{4-}$ and DFe concentrations were observed after wet dust deposition using high amounts of dust previously leached in low pH rain ($H_2SO_4$, pH=2). Remarkably, the experiments showed no nutrient release ($PO_4^{3-}$, $SiO_4^{4-}$ and DFe) from dry-dust addition and the $NO_3^-$ concentrations remained unaffected in all (dry and wet) experiments. The prokaryotic cyanobacterium *Synechococcus* spp. was the most prominent picophytoplankton in all mixed layer experiments. After an initial increase in cell abundance, a subsequent decrease (at M1) or a slight increase (at M3) with similar temporal dynamics was observed for dry- and wet-dust deposition experiments. The POC concentrations increased in all experiments and showed similar high values after both dry and wet dust deposition treatments, even though wet dust deposition is considered to have a higher potential to introduce bioavailable nutrients (i.e. $PO_4^{3-}$, $SiO_4^{4-}$ and DFe) into the otherwise nutrient-starved oligotrophic ocean. Our observations suggest that such nutrients may be more likely to favor the growth of the phytoplankton community when an additional N-source is also available. In addition to acting as a fertilizer, our results from both dry and wet dust deposition experiments suggest that Saharan dust particles might be incorporated into marine snow aggregates leading to similar high POC concentrations.

*Keywords:* Saharan dust, incubation, nutrients, POC, deposition, phytoplankton, ocean fertilization



## 1 Introduction

Atmospheric dust deposition is one of the major processes to deliver nutrients to the oligotrophic regions of the global ocean where river runoff and upwelling are either absent or not strong enough to sustain high levels of phytoplankton productivity. Massive amounts of mineral dust (182 Tg year$^{-1}$) leave the coast of northern Africa (Yu et al., 2015) to be

transported further west by the trade winds. During transport, a great part of this dust is deposited into the ocean by dry deposition through gravitation, or by wet deposition with rain. Saharan dust deposition has been shown to have a great impact in the biogeochemical cycling of oceanic waters underneath the most prominent dust plumes blown towards the Mediterranean Sea (Bonnet et al., 2005; Romero et al., 2011; Desboeufs et al., 2014; Guieu et al., 2014a; Ridame et al., 2014) and towards the tropical North Atlantic Ocean (Jickells, 1999; Baker et al., 2003; Mills et al., 2004; Bristow et al., 2010). Both fertilization

(Jickells et al., 2005; Pabortsava et al., 2017; Guerreiro et al., 2017) and ballasting of marine snow aggregates (Van der Jagt et al., 2018) from dust deposition impact the carbon dioxide content of surface waters. Mesocosm experiments in the Mediterranean Sea revealed that dust predominates the particulate phase that is exported to the base of the mesocosms (Desboeufs et al., 2014) and that up to 50 % of the particulate organic carbon flux (POC) can be associated with lithogenic dust particles (Bressac et al., 2014). Wet-dust deposition results in a higher export production of POC compared to dry-dust

deposition (Desboeufs et al., 2014), likely due to stimulated chlorophyll-$a$ production (Ridame et al., 2014). This is attributed to cloud processing, which is responsible for 'aging' the dust minerals during their long-range transport while being exposed to acidic conditions in the atmosphere (Desboeufs et al., 2001). Thereby, the atmospheric pH is a major factor regulating metal solubility can be as low as 1 or 2 in atmospheric aerosols and clouds, as a result of anthropogenic $H_2SO_4$ and $HNO_3$ uptake as well as $SO_2$ oxidation (Spokes et al., 1994; Shi et al., 2011). These low pH conditions are especially favorable for leaching

iron from the dust particles into the rain droplets, and as a result iron is more easily released into the ocean waters (Spokes et al., 1994). Iron is an important micronutrient needed for phytoplankton growth and nitrogen fixation (Falkowski et al., 1998; Capone, 2001; Karl et al., 2002), especially in the so-called High Nutrient Low Chlorophyll (HNLC) regions of the global ocean. In regions where phytoplankton is not limited by iron but by phosphorous and nitrogen, such as the tropical Atlantic Ocean and the Mediterranean Sea, Saharan dust is important in promoting nitrogen-fixation which in turn enhances new

production in the surface ocean (Ridame and Guieu, 2002; Baker et al., 2003), by delivering nitrogen compounds to other phytoplankton groups. By means of bioassay experiments, Mills et al. (2004) showed that diazotrophic nitrogen-fixation in the eastern tropical North Atlantic Ocean is co-limited by iron and phosphorous and that Saharan dust is likely to relieve such co-limitation. Mahaffey et al. (2003) also concluded that nitrogen-fixation in the North Atlantic Ocean is temporally relieved by episodic Saharan dust deposition. Moreover, Baker et al. (2013) demonstrated that total iron input is higher when

precipitation rates are also high. Still, phosphorus' availability for phytoplankton growth across the tropical North Atlantic Ocean is low (Sohm et al., 2011) as strong vertical water stratification reduces the introduction of nutrients from the deep into the upper mixed layer. Furthermore, as the nutricline deepens westward across the tropical North Atlantic (Guerreiro et al., 2017), phytoplankton production in the upper layers of the open ocean may depend on atmospheric dust deposition delivering the necessary nutrients. Based on an inverse correlation between low nitrogen isotopic composition and the POC flux in the

North oligotrophic gyre, Pabortsava et al. (2017) showed a link between POC fluxes and newly fixed nitrogen by diazotrophs whose activity was interpreted to be stimulated by Saharan dust driven nutrient input. However, to what extent Saharan dust deposition is responsible for new production of POC in the Atlantic Ocean, and whether dry- and wet-dust deposition have different impacts remains difficult to assess due to the complex interplay between dust sources, atmospheric processing, depositional processes, timing of deposition and surface-water conditions.

Here we examine the potential of Saharan dust as a fertilizer for on phytoplankton growth and POC production in the Atlantic Ocean, underneath the most prominent dust plume at 12°N (Mulitza et al., 2008). For our experiment, we incubated two different types of Saharan dust mimicking dry- and wet-dust deposition to the ocean and quantified the nutrient release





(e.g. phosphate, $PO_4^{3-}$, silicate, $SiO_4^{4-}$ and nitrogen, $NO_3^-$) and its effect on picoplankton cell abundances during the experiment. Furthermore, we estimated the effect of dust additions on POC, dissolved inorganic carbon (DIC) as well as dissolved iron (DFe) at the start and end of the experiments. The results of this study contribute to a better understanding of the biogeochemistry in the equatorial North Atlantic Ocean and reveal the importance of Saharan dust deposition on the
productivity in the surface waters.

## 2 Material and Methods

### 2.1 Background information

Saharan-dust incubations were conducted along a 12°N transect defined by three stations (M1, M3 and M4, Fig. 1) while crossing the tropical North Atlantic during March-April 2016 on board the research vessel RRS *James Cook,* cruise *JC*
*134* (Stuut et al., 2016). Saharan dust had been sampled earlier along this transect using sediment traps to determine downward particles fluxes as well as Saharan dust particle sizes (Van der Does et al., 2016; Korte et al., 2017). According to these studies dust particles collected closest to the African coast revealed to be larger and more similar to the dust emitted from the Sahara Desert. With increasing distance to the source, the mineral and chemical composition of the deposited dust gradually changed due to preferential settling of coarser dust particles (i.e. quartz) in closer proximity to the sources (Korte et al., 2017). Particle
sizes of the collected dust showed a decrease in modal diameter towards the west from about 18 µm at M1 to about 10 µm at M4 (on an annual basis; Van der Does et al., 2016).

Two contrasting dust types were used for the incubation experiments that were collected from the foothills of the Atlas mountain (Fig. 1), located in a potential dust source and dust transit area in the western coastal region of Mauritania (Scheuvens et al., 2013). The first dust type was taken from an old lake deposit (lake dust), expected to contain more
bioavailable nutrients associated with fine-grained clay sediments (Scheuvens et al., 2013;Bristow et al., 2010), and the second dust type was taken from a sand dune (dune dust), consisting of coarser-grained sediments.

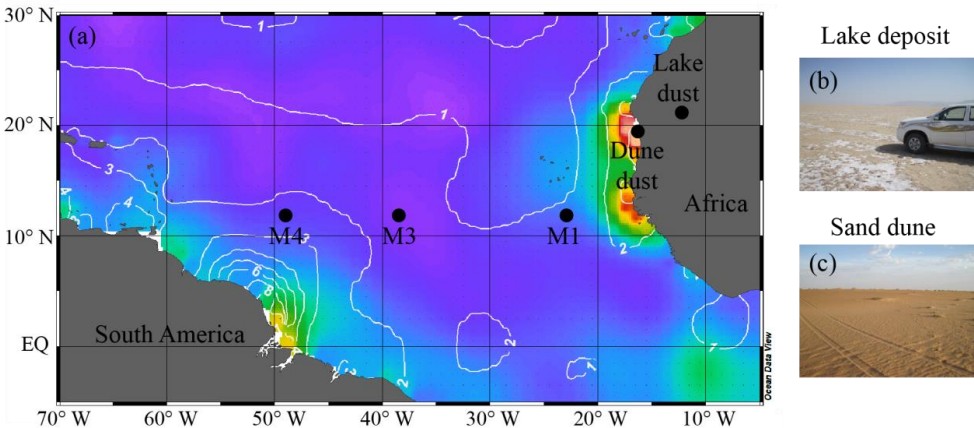

**Figure 1. (a) Map showing the sites at which dust incubation experiments were carried out in the tropical North Atlantic Ocean (M1 – M4). The color shading displays phosphate concentrations with values increasing from blue = 0 to red =**
**0.6 µmol L⁻¹, and white lines represent silicate concentrations (µmol L⁻¹) in the surface waters (Garcia et al., 2014). For the experiments, dust samples were collected at two sites in Mauritania, from (b) a lake deposit (21°N, 12°W), and (c) a sand dune field (19°N, 16°W).**





### 2.2 Dust characteristics

The grains of the collected sand and lake soil sediments from Mauritania were too coarse to represent an atmospheric dust sample, showing a bimodal (sand) and multimodal (lake soil) distribution with modal grain sizes at 1, 100, 200 and 900 µm, while dust deposited at site M1 in March 2013 is much finer grained with a unimodal distribution peaking at 18 µm (van

der Does et al. 2016; Fig. 2a). Therefore, in order to mimic a dust sample typically deposited in this location, the sand and lake soil sediments were sieved and sorted into different size classes (< 32, < 63, < 90, < 150, < 250 µm). For both dust types, the < 32 µm size fraction was used for the incubation experiments, which was ground for the dune dust, but not for the lake dust (Fig. 2b).

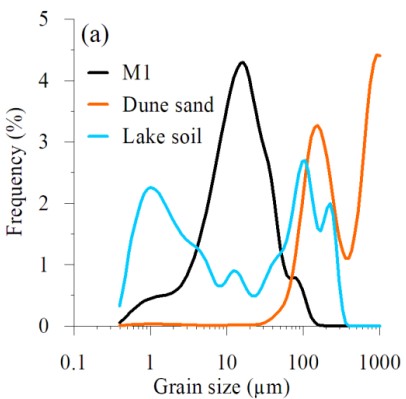
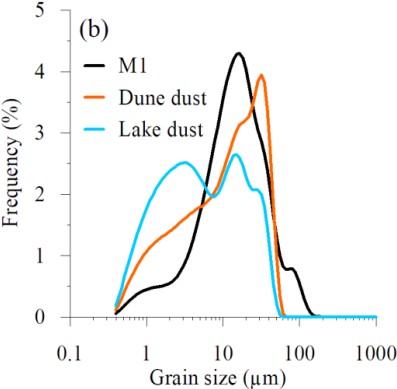

**Figure 2. Grain-size distributions of dune and lake sediments (a) before and (b) after sieving. Black line represents Saharan dust from a sediment trap at 1200 m water depth at site M1 deposited in March 2013 (Van der Does et al., 2016), the month in which the incubations took place.**

### 2.3 Incubation experiments

Before usage on board, all incubation bottles (6 L, PET) were cleaned with 10% HCl, rinsed with Milli-Q, dried, and sealed in plastic bags. Water sampling was performed with a Seabird CTD equipped with a rosette holding 24 Niskin bottles of 10 L. Seawater was sampled from the upper mixed layer (ML, ~20 m) at stations M1, M3 and M4. Additional water sampling was taken from the deeper mixed layer (DL) shortly above the deep chlorophyll maximum at ~72 m at M3, where temperatures

were still similar to the surface water. At M4, seawater was also taken from 100 m depth. All temperature and fluorescence profiles are shown in supplementary Fig. S1.

In a laminar flow hood (ISO class 5) the seawater-filled incubation bottles were amended with the dune or lake dust, which were either acidified, mimicking wet-dust deposition or kept untreated for testing the effect of dry-dust deposition. Control samples with no dust addition and/or addition of acidified "rain" only (rain-control) were run along the incubations. Dust-

addition treatments, their duration and the water sampling depths are given in Table 1 for each of the three incubation sites (M1, M3 and M4).

Attending to the fine-grained nature of the lake dust sediments and their associated expectedly higher nutrient content (Bristow et al., 2010), we tested the impact of dry versus wet deposition of dust into seawater with the lake dust, while for the dune dust only the effects of wet deposition were tested. To mimic wet deposition, both dust types were leached in 40 mL

'artificial rainwater' for 16 to 24 hours before adding the treatment to the incubation bottles. This timespan was chosen according to protocols from previous cloud processing experiments (Spokes et al., 1994; Shi et al., 2009). Based on satellite-

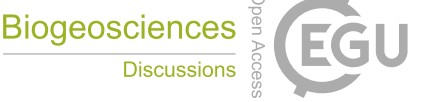



derived precipitation data, 40 mL translates into a realistic precipitation rate (0.04 mm d$^{-1}$, Fig. S2) for our incubation sites during the time of the experiment. According to Van der Does et al. (2016), a small amount of precipitation is already sufficient to wash out suspended dust from the atmosphere by wet deposition.

**Table 1. Overview of incubation dust treatments at stations M1, M3 and M4. N = number of replicates. ML= mixed layer, DL = deeper mixed layer. *results of incubations at M4 are given in the supplement S4.**

| Site | Depth | Duration (days) | Treatment | | | |
|---|---|---|---|---|---|---|
| | | | Control | Wet deposition | | Dry deposition |
| **M1 (23°W)** | ML (20m) | 8 | Control (N=3) Rain-control (N=3) | Lake dust 0.25 mg, pH 2 (N=3), 1.5 mg, pH 2 (N=3) | Dune dust 0.25 mg, pH 2 (N=3), 1.5 mg, pH 2 (N=3) | Lake dust 0.25 mg (N=3), 1.5 mg (N=3) |
| **M3 (38°W)** | ML (20m) | 4 | Control (N=6) | Lake dust 0.25 mg, pH 2 (N=3) | | |
| | DL (72m) | 4 | Control (N=6) | Lake dust 0.25 mg, pH 2 (N=3), 0.25 mg, pH 4.5 (N=3) | | |
| **M4 (49°W)*** | ML (20m) | 4 | Control (N=4) | Lake dust 0.25 mg, pH 2 (N=3), 0.25 mg, pH 4.5 (N=3) | Dune dust 0.25 mg, pH 2 (N=3), 0.25 mg, pH 4.5 (N=3) | Lake dust 0.25 mg (N=3) |
| | DL (100m) | 4 | Control (N=3) | Lake dust 0.25 mg, pH 2 (N=3) 0.25 mg, pH 4.5 (N=3) | Dune dust 0.25 mg, pH 2 (N=3), 0.25 mg, pH 4.5 (N=3) | Lake dust 0.25 mg (N=3) |

The artificial rain was prepared from Milli-Q water amended with sulfuric acid ($H_2SO_4$), which lowered the pH to either 2 or 4.5 (Table 1), mimicking atmospheric conditions and natural rain pH, respectively (Spokes et al., 1994). $H_2SO_4$ was used as it is known as an anthropogenic acid component in cloud water (Jickells et al., 1982; Desboeufs et al., 2001). To prevent the release of any nitrogen other than Saharan dust during the experiments, other atmospheric acids, such as $HNO_3$, were avoided. To test the impact of a low and high dust deposition event, we performed two separate experiments adding different amounts of dust, i.e., 0.25 and 1.5 mg L$^{-1}$ (Table 1). While for the wet-deposition treatment, the dust was added to incubation bottles together with the rain, the dust for the dry deposition treatment was added directly to the incubation bottles by rinsing the dust-containing vial with ambient seawater.

After dust addition, the incubation bottles were closed, sealed and hooked up-side down onto wooden trays at the bottom of the on-deck incubation tubs to expose them to natural light conditions. Using natural screens and meshes, ambient light was adjusted to the actual irradiance (PAR, Table S1) varying between 220 and 7 W m$^{-2}$ (M4, 20 and 100 m, respectively), 88 and 14 W m$^{-2}$ (M3, 20 and 72 m, respectively), and 25 W m$^{-2}$ (M1, 20 m). The light intensity in the incubators was measured by a Li-Cor PAR meter. Circulating surface seawater surrounded the incubation bottles within a temperature range of ± 2°C from the original sampling site. From day 0 (immediately after dust addition) or day 1 until day 4 or day 8, daily subsamples were taken from each incubation bottle for analyzing the macronutrient concentrations ($PO_4^{3-}$, $SiO_4^{4-}$, $NO_3^-$) and flow cytometry analyses. During transportation from the incubators to the ship's laboratory for subsampling, the incubation bottles were placed in darkened plastic bags shielding them from direct sunlight. Subsampling was carried out inside a laminar flow hood to avoid contamination. At the start and end of the incubation, samples for dissolved iron (DFe) and dissolved inorganic carbon (DIC) were also taken. Samples for the analysis of particulate organic carbon (POC) were taken only at the end of the experiments. The original CTD water composition served as baseline for all parameters of the incubations.





The water-handling procedures for incubations improved from experience gained during the first incubations in the west (M4). At M4, all sampled seawater was mixed in a darkened acid-cleaned 240 L tank, whereas at M3 and M1, the seawater was directly and gently filled into the darkened incubation bottles, which shortened the incubation procedure. Due to the different handling at station M4, results of the incubations from this location are presented in the supplement (S4).

### 2.4 Analytical measurements

Samples for dissolved inorganic macronutrient concentrations were filtered through a 0.2 µm Acrodisc filter and stored either frozen at -20 °C for $NO_3^-$ and $PO_4^{3-}$ or cooled at +4 °C for $SiO_4^{4-}$. All samples were analyzed on a TRAACS Gas Segmented Continuous Flow Analyser at NIOZ using the colorimetric methods (Murphy and Riley, 1962; Strickland and Parsons, 1972; Grasshoff et al., 1983). Filters (25 mm GF/F) for POC measurements were muffled at 450°C for 5 hours and pre-weighed to determine the sample amount after filtration. After sample filtration on board, POC filters were stored frozen (-20° C) and later analyzed using a Thermo-Interscience Flash EA112 Series NC analyzer at NIOZ. Therefore, filters were freeze dried, weighed again and acid fumed to dissolve inorganic carbon. Subsequently, filters were folded into tin capsules, analyzed against the certified Acetenalide standard and blank corrected. Flow cytometry cell counts were performed on board the ship on fresh phytoplankton samples using a BD Accuri 6 flow cytometer with a 20mW solid state blue laser (488 nm excitation). The trigger was set on red chlorophyll-*a* autofluorescence and the different phytoplankton populations were discriminated based on their red and orange phycoerythrin autofluorescence (e.g. indicative for the cyanobacteria *Synechococcus* spp.) and scatter signal. Data analysis was performed using FCSExpress5 software. Samples for DIC measurements were fixed with 10 µL of concentrated $HgCl_2$ and stored at room temperature until measured at NIOZ after the method of Stoll et al. (2001). Statistical analysis was done with Sigmaplot one-way Anova, followed by a Bonferroni t-test when a statistically significant difference was found ($p < 0.001$).

### 3 Results

### 3.1 Characterization of the original seawater sampled at M1, 23°W

The original seawater taken from 20 m depth showed temperatures of 24° C (Table S1) and low concentrations of the macronutrients $PO_4^{3-}$ (0.03 µmol $L^{-1}$), $SiO_4^{4-}$ (0.71 µmol $L^{-1}$), and $NO_3^-$ (0.02 µmol $L^{-1}$). The original trace metal concentration of DFe was 1.0 nmol $L^{-1}$, while POC and DIC showed concentrations of 0.04 mg $L^{-1}$ and 2125 µmol $L^{-1}$, respectively. At the start of the experiment, the water contained an abundance of $1.6 \cdot 10^4$ *Synechococcus* spp. cells per mL. The lake dust was added to the original seawater as dry and wet deposition, while the dune dust was only added as wet deposition to the incubations (Table 1).

### 3.1.1 Nutrient development and organic material changes after wet and dry dust additions at M1

*$PO_4^{3-}$ concentration:* Large differences in $PO_4^{3-}$ concentrations were observed between the dry and wet lake dust deposition treatments (Fig. 3a and b). The $PO_4^{3-}$ concentrations of both low (0.25 mg $L^{-1}$) and high (1.5 mg $L^{-1}$) dry lake dust deposition treatments were similar in comparison to the control sample and remained as low as in the original seawater (~0.03 µmol $L^{-1}$) throughout the experiment (Fig. 3a). Also, after adding the low amount of wet lake dust, the $PO_4^{3-}$ concentrations remained comparably low (Fig. 3b) Only after adding the high amount of wet lake dust, $PO_4^{3-}$ concentrations doubled immediately after dust addition at day 0 to 0.06 µmol $L^{-1}$ and showed a gradual decrease to ~0.03 µmol $L^{-1}$ during the following eight incubation days (Fig. 3b).

The $PO_4^{3-}$ concentration of the low wet dune dust deposition ranged between 0.03 and 0.04 µmol $L^{-1}$ after dust addition until the end of the experiment (Fig. 3c) and was therefore comparable to the original seawater (0.03 µmol $L^{-1}$) and the low lake dust deposition treatment (~0.03 µmol $L^{-1}$, Fig. 3b). In contrast, the $PO_4^{3-}$ concentration of the high wet dune dust





deposition doubled to 0.06 µmol L$^{-1}$ immediately after dust addition at day 0, similar to the lake dust, and decreased to 0.05 µmol L$^{-1}$ at the end of the experiment.

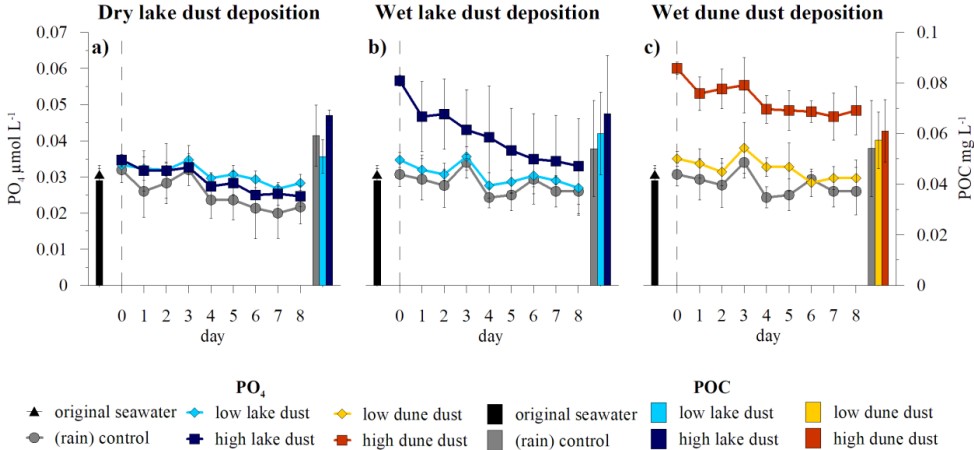

**Figure 3. Phosphate (lines, left axis) and particulate organic carbon (bars, right axis) concentrations with error bars showing triplicate standard deviation during incubations for a) dry and b) wet lake dust deposition, and c) wet dune dust deposition treatments at M1. Dashed line represents start of the experiment.**

*POC concentration:* While the original POC concentration of the CTD seawater was 0.04 mg L$^{-1}$, concentrations
increased to values ranging between 0.05 and 0.07 mg POC L$^{-1}$ at the end of all experiments for dry and wet lake dust deposition with low and high dust amounts (Fig. 3). However, the increase in POC was only significantly higher from the original values for the high dry lake dust deposition treatment (Fig. 3a).

The average POC concentrations for the dune dust deposition treatments ended in a small increase in comparison to the original value of 0.04 mg L$^{-1}$ (Fig. 3c). The POC values of the rain-control treatment at the end of the experiment showed
the smallest increase, followed by the low- and the high wet dune dust deposition treatments. Average concentrations of the three treatments ranged between 0.05 and 0.06 mg L$^{-1}$. These values are in the same order of magnitude as the resulting POC concentrations of the wet lake dust deposition treatments.

*SiO$_4^{4-}$ concentration.* The SiO$_4^{4-}$ concentration in both high and low dry lake dust deposition treatments resulted in similar values as found in the original seawater and the control sample (~0.71 µmol L$^{-1}$, Fig. 4a). The SiO$_4^{4-}$ concentration for
both low and high wet lake deposition treatments, however, resulted in higher concentrations (0.88 and 1.55 µmol L$^{-1}$, respectively) in comparison to the original seawater and the rain-control sample immediately after dust addition at day 0 (Fig. 4b). The SiO$_4^{4-}$ concentrations increased with the high wet lake deposition treatment during the experiment, while it slightly decreased in the low wet lake deposition treatment or remained unchanged in the rain-control sample.

The SiO$_4^{4-}$ concentration of the dune dust deposition treatments was lower than the SiO$_4^{4-}$ concentrations of the lake
dust deposition but showed similar temporal patterns. The SiO$_4^{4-}$ concentration of the high dune dust deposition treatment was clearly offset (0.96 µmol L$^{-1}$) compared to the original seawater (0.71 µmol L$^{-1}$) and the rain-control sample (0.70 µmol L$^{-1}$) immediately after dust addition at day 0. At day 1, SiO$_4^{4-}$ concentrations increased even more to 1.08 µmol L$^{-1}$ and subsequently stayed high. The offset of the low dune dust treatment remained small (Fig. 4c).



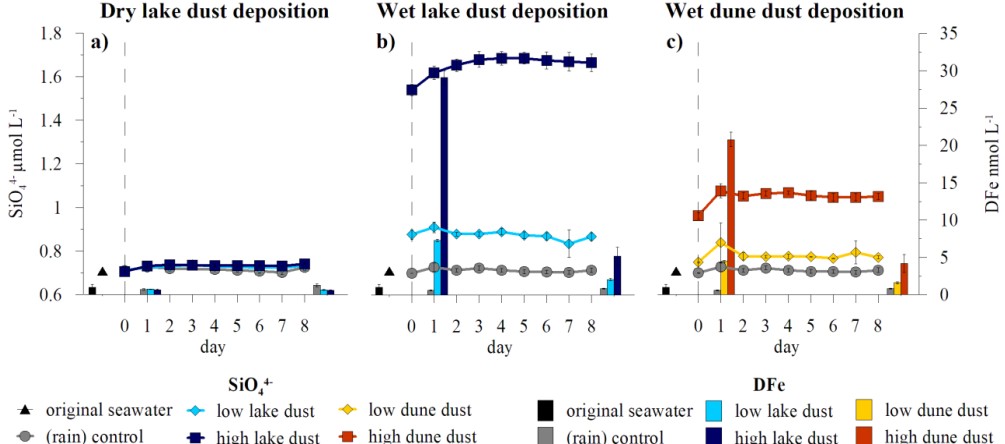

**Figure 4. Temporal dynamics of dissolved silicate (lines, left axis) and dissolved iron concentration (bars, right axis) with error bars showing standard deviation of triplicate measurements for incubations at M1 for a) dry and b) wet lake dust deposition and c) wet dune dust deposition treatments. Dashed line represents start of the experiment.**

*DFe concentration:* While the DFe concentrations with both high and low dry lake dust deposition treatments remained as low as the original DFe concentration (0.8 nmol $L^{-1}$ and 1.0 nmol $L^{-1}$, respectively) at day 1, they increased strongly after low and high wet lake dust deposition (7 and 30 nmol $L^{-1}$, respectively). Towards the end of the experiment, DFe concentrations decreased to 2 and 5 nmol $L^{-1}$ for the low and high wet lake dust deposition treatments, respectively, while

staying low at 0.6 nmol $L^{-1}$ in both dry lake dust deposition treatments (Fig. 4a & b). Only the control sample had slightly higher DFe concentrations (1.2 nmol $L^{-1}$) at day 8, potentially caused by contamination during analysis. The pure rain (pH=2) had no effect on the DFe concentration of the seawater since DFe concentrations stayed low throughout the experiment and no differences in DFe concentrations were seen between both control samples (dry and wet).

The DFe concentrations of the wet dune dust deposition also showed a similar behavior as seen after wet lake dust

deposition. While DFe concentration of the high wet dune dust deposition was 20 nmol $L^{-1}$ at day 1 with a significant decrease to 4 nmol $L^{-1}$ at day 8, the low wet dune dust deposition treatment started with a DFe value of around 4 nmol $L^{-1}$ at day 1 and decreased to 1.6 nmol $L^{-1}$ at day 8 (Fig. 4c).

*$NO_3^-$ concentration:* The $NO_3^-$ concentrations showed irregular peaks throughout the experiment for dry and wet deposition of the lake dust (Fig. 5a & b), with no differences between both deposition types or to the control samples.

Concentrations ranged between 0 and 1 μmol $L^{-1}$ in all treatments and showed large error bars pointing to inhomogeneity in the three replicates. The results show that there is no $NO_3^-$ input from the lake dust.

The $NO_3^-$ concentrations of the dune dust stayed as low as the $NO_3^-$ concentrations of the wet lake dust treatments and showed no difference with the control sample showing that our processed dust did not deliver nitrate to the seawater. During the 8 days of the experiment, concentrations ranged between 0 and 1.2 μmol $L^{-1}$ with irregular peaks and large error

bars (Fig. 5c).





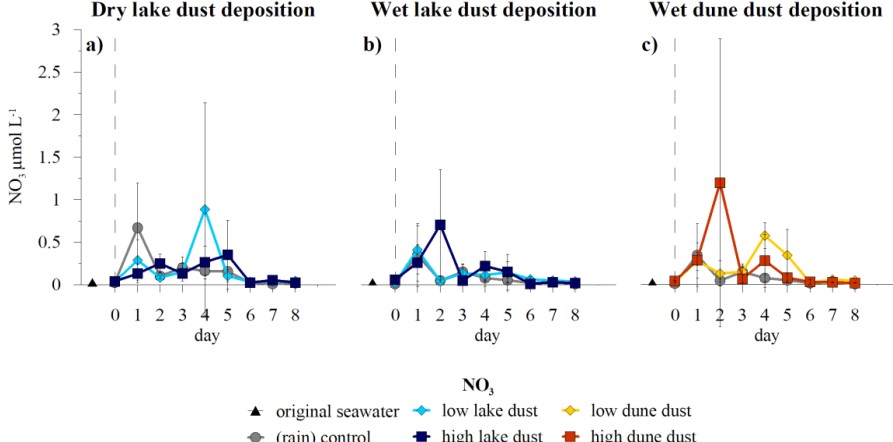

**Figure 5. Temporal dynamics of nitrate concentrations during a) dry and b) wet lake dust deposition and c) wet dune dust deposition at site M1. Error bars show standard deviation of triplicate measurements and peaks are attributed to contamination. Dashed line represents start of the experiment.**

*Synechococcus* spp*.:* The prokaryotic cyanobacterium *Synechococcus* spp. was by far the dominant picophytoplankton group ranging between 45 and 84 % of the total cell counts in both the dry and wet dust deposition treatments (Fig. 6). *Synechococcus* spp. abundance accounted for $1.6 \cdot 10^4$ cells mL$^{-1}$ in the original seawater and in all incubation bottles immediately after dust addition at day 0. After the first day of incubation cell abundances doubled to $3.3 \cdot 10^4$

10   mL$^{-1}$ in all treatments with no significant difference between dry and wet dust deposition (Fig. 6a & b). Between day 1 and 8, cell counts for both dry lake dust deposition treatments decreased to ~$0.7 \cdot 10^4$ mL$^{-1}$. Cell counts for the wet lake dust deposition treatments also decreased from $3.3 \cdot 10^4$ mL$^{-1}$ to $0.7 \cdot 10^4$ mL$^{-1}$, albeit, more slowly after high dust addition, as observed from the relatively high cell counts until day 5 (Fig. 6b).

Cell counts of the cyanobacterium *Synechococcus* spp. of the wet dune dust deposition treatments were slightly lower

15   than for the lake dust deposition treatments but displayed a similar pattern with a significant increase from $1.6 \cdot 10^4$ up to $3.0 \cdot 10^4$ cells mL$^{-1}$ after the first day, and a subsequent decrease to $0.5 \cdot 10^4$ cells mL$^{-1}$ at day 8 (Fig. 6c).

*DIC concentration:* The original DIC concentrations differed only slightly from the end concentrations and are therefore considered to remain unaltered for dry and wet lake dust treatments (Fig. 6).

The DIC concentrations at the end of the wet dune dust deposition treatments also only decreased slightly in

20   comparison to the original seawater, with the strongest decrease observed after the high dune dust deposition (Fig. 6c).



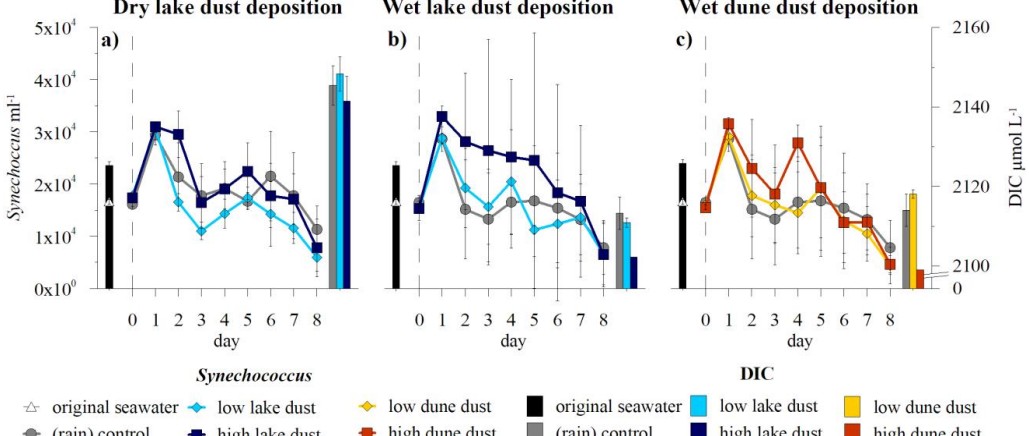

**Figure 6. Temporal dynamics of *Synechococcus* spp. abundance (lines, left axis) and dissolved inorganic carbon concentration (DIC, bars, right axis) with error bars showing standard deviation of triplicate measurements for incubations at M1 for (a) dry and (b) wet lake dust deposition and (c) wet dune dust deposition. Dashed line represents start of the experiment.**

### 3.2 Characterization of the original seawater sampled in the upper mixed layer (ML; 20 m) at M3, 38°W

The mixed layer temperature at 20 m water depth was 25° C (Table S1). The original macronutrient- and trace metal concentrations in the ML were < 0.02 µmol L$^{-1}$ for PO$_4^{3-}$, 0.82 µmol L$^{-1}$ for SiO$_4^{4-}$, 0.14 µmol L$^{-1}$ for NO$_3^-$ and < 0.5 nmol L$^{-1}$ for DFe (Fig. 7a-d). Original abundance of *Synechococcus* spp. cells accounted for $1.8 \cdot 10^3$ cells per mL (Fig. 7e). In addition, a second picoeukaryotic phytoplankton group with a starting abundance of $0.4 \cdot 10^3$ cells per mL was identified at M3 (Fig. 7f). The overall POC concentration was 0.04 mg L$^{-1}$, while DIC concentration was 2130 µmol L$^{-1}$ (Fig. 7g and h).

#### 3.2.1 Nutrient development and organic material changes after wet dust deposition in the ML

At M3, wet lake dust (0.25 mg L$^{-1}$, leached in pH 2 rain) was added to the seawater taken at 20 m water depth (Table 1). The progression of nutrient concentrations after adding the wet dust were similar to the nutrient progression at M1 with the same wet dust addition. The PO$_4^{3-}$ concentrations stayed low until the end of the experiment (~0.02 µmol L$^{-1}$) similar to the original seawater and the control sample (Fig. 7a). The SiO$_4^{4-}$ concentrations increased to 1.0 µmol L$^{-1}$ at day 1 after dust addition and stayed at this elevated level (Fig. 7b). The offset of 0.18 µmol L$^{-1}$ compared to the original seawater is the same as for the offset observed for the SiO$_4^{4-}$ increase at station M1. The SiO$_4^{4-}$ concentrations of the control sample remained low at 0.84 µmol L$^{-1}$. The NO$_3^-$ concentrations showed varying values for the three replicates resulting in large error bars. Still, average values showed an increase from day 1 to day 3 of 0.04 to 0.63 µmol L$^{-1}$ and a subsequent decrease to 0.04 µmol L$^{-1}$ after wet dust deposition (Fig. 7c). The control sample however also showed a NO$_3^-$ increase from 0.04 to 1.2 µmol L$^{-1}$ at day 2 and a decrease to 0.02 µmol L$^{-1}$ towards the end of the experiment. The DFe concentration increased to > 4 nmol L$^{-1}$ after dust addition at day 1 and decreased to 1.5 nmol L$^{-1}$ at the end of the experiment, while DFe concentrations remained unchanged in the control sample (Fig. 7d). The cyanobacterium *Synechoccocus* spp. abundance was about one magnitude lower than at station M1 but showed an increase from the original seawater to a constant cell abundance of around $3 \cdot 10^3$ cells mL$^{-1}$ throughout the experiment (Fig. 7e), although not different from the control sample. The other picoeukaryotic phytoplankton group (*R3*), however, did show differences between the dust addition and the control sample. While cell abundances in the treated sample and the control sample were similar to the original seawater at day 1, they started to diverge from day 3 leading to a higher cell abundance of $1.5 \cdot 10^3$ after wet dust addition at day 4 (Fig. 7f), a significant difference to the original seawater. Nevertheless, there was no change in the bulk POC concentrations after dust addition. The POC



concentrations remained with 0.04 mg L$^{-1}$ after dust addition as low as in the original seawater, while the control sample accounted for similar values (Fig. 7g). The DIC concentrations decreased to 2110 and 2075 µmol L$^{-1}$ in the control sample and dust addition treatment (Fig. 7h).

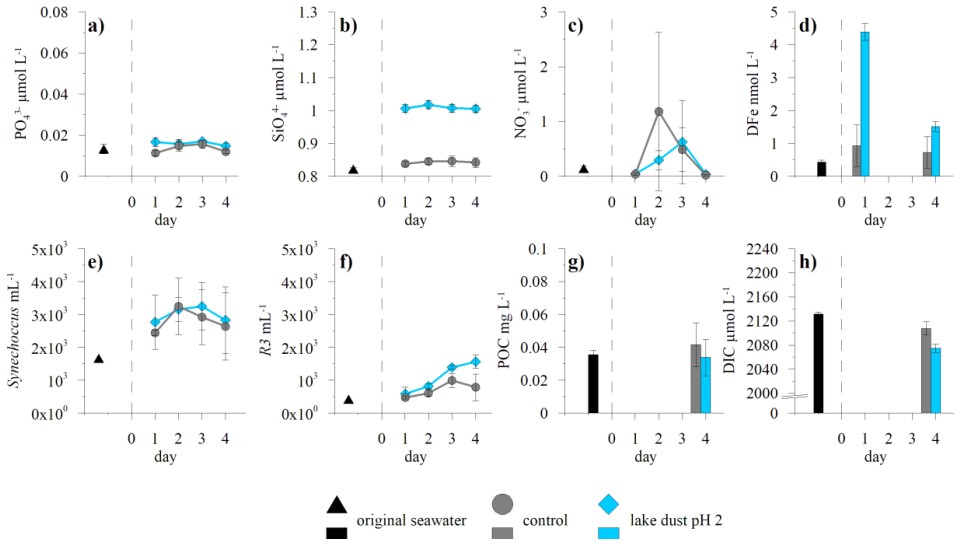

**Figure 7. Temporal dynamics of nutrient concentrations in the seawater from the ML at 20 m for a) phosphate, b) silicate, c) nitrate, and d) dissolved iron. Picoplankton cell abundances for e) *Synechoccus* spp and f) an unidentified picoeukaryotic phytoplankton group (*R3*). Concentrations for g) particulate organic carbon and h) dissolved inorganic carbon. Error bars show standard deviation of triplicate measurements.**

**3.3 Characterization of the original seawater sampled in the deeper mixed layer (DL; 72 m) at M3, 38°W**

The CTD profiles showed uniform temperature profiles of around 25°C from the surface down to 73 m water depth (Table S1). Fluorescence increased in depth, reaching a maximum at 105 m, signaling the presence of the DCM (Fig. S1b). Water for the incubation was taken from 72 m water depth where temperatures were similar to the surface. However, at this depth the nutrient concentrations are already increased. The original macronutrient- and trace metal concentrations in the DL (72 m) were 0.07 µmol L$^{-1}$ for PO$_4^{3-}$, 1.04 µmol L$^{-1}$ for SiO$_4^{4-}$, 0.65 µmol L$^{-1}$ for NO$_3^-$, and 0.7 nmol L$^{-1}$ for DFe (Fig. 8a-d). The original cell abundance of *Synechoccus* spp. was only 0.4 · 10$^3$ cells per mL and therefore as low as in the ML (Fig. 8e). The second picoeukaryotic phytoplankton group (*R3*) however, showed a higher abundance of 2.7 · 10$^3$ cells per mL in the DL (Fig. 8f). Still, the POC concentrations were lower in the DL and showed values of 0.02 mg L$^{-1}$ (Fig. 8g)., The DIC concentrations of 2200 µmol L$^{-1}$were slightly higher than in the ML (Fig. 8h).

**3.3.1 Nutrient development and organic material changes after wet dust deposition in the DL**

Wet lake dust (0.25 mg L$^{-1}$), leached in pH 2 and pH 4.5 rain, was added to the seawater sampled in the deeper mixed layer at 72 m water depth (Table 1). For the dust leached in pH 2 rain, the same pattern of the nutrient development as seen in the ML was observed. While there was no change in the PO$_4^{3-}$ concentration in comparison to the original seawater and the control sample (Fig. 8a), there was a clear offset to the original seawater and the control sample in SiO$_4^{4-}$ concentration after dust addition (Fig. 8b). However, both the PO$_4^{3-}$ and SiO$_4^{4-}$ concentrations decreased during the course of the experiment. While the PO$_4^{3-}$ concentrations decreased from 0.08 µmol L$^{-1}$ to 0.05 µmol L$^{-1}$ between day 1 after dust addition and day 4, the SiO$_4^{4-}$ concentrations decreased from 1.2 µmol L$^{-1}$ to 1.1 µmol L$^{-1}$ at the same time. The NO$_3^-$ concentrations did not

change with dust addition in comparison to the original seawater and decreased from 0.6 µmol L⁻¹ to 0.2 µmol L⁻¹ from day 1 to day 4 (Fig. 8c). The DFe concentrations showed a strong increase to > 4 nmol L⁻¹ at day 1 after dust addition and a decreased to around 2.5 nmol L⁻¹ towards the end (Fig. 8d). The picoplankton cells showed the same progression with and without dust addition, while the *R3* group showed highest counts in the control sample (Fig. 8f). A significant change in POC was observed

5   in comparison to the 0.02 mg L⁻¹ of the original seawater. The concentrations increased to ~0.07 mg L⁻¹ at the end of the experiment for both dust additions as well as the control sample (Fig. 8g). The DIC concentration decreased from the concentration of the original seawater of 2200 µmol L⁻¹ to 2120 µmol L⁻¹ (Fig. 8h).

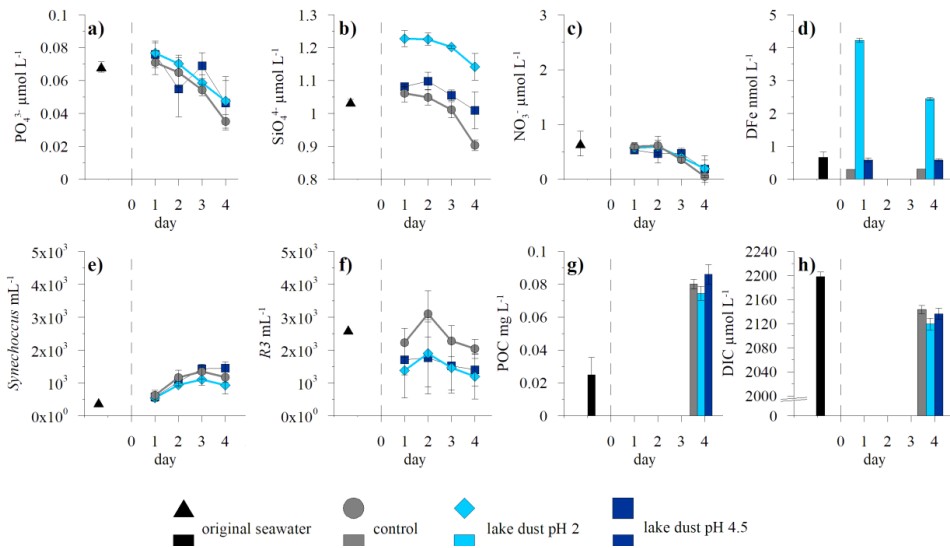

**Figure 8. Temporal dynamics of nutrient concentrations in the seawater from the DL at 72 m for a) phosphate, b) silicate, c) nitrate, and d) dissolved iron. Picoplankton cell abundances for e)** *Synechoccus* **spp and f) an unidentified picoeukaryotic phytoplankton group (*R3*). Concentrations for g) particulate organic carbon and h) dissolved inorganic carbon. Error bars show standard deviation of triplicate measurements.**

The addition in pH 4.5 rain did not affect the seawater in terms of nutrient release or abundance in picoplankton cells. The macronutrients $PO_4^{3-}$, $SiO_4^{4-}$, and $NO_3^-$, decreased during the course of the experiment in the same way as observed in the control sample (Fig. 8a-e). No DFe was released into the seawater when dust in pH 4.5 rain was added as the values remained at 0.6 nmol L⁻¹, similar to the concentration observed in the original seawater (Fig. 8d). The picoplankton cell abundances (Fig.

20   8e and f) were similar for dust in pH 4.5 and pH 2 rain but showed a slight increase throughout the experiment in the DL for *Synechoccus* spp. (Fig. 8e). The POC concentration, however, showed a four-fold increase at the end of the experiment to 0.09 mg L⁻¹ compared to the original seawater (Fig. 8g). The DIC concentration decreased from the original seawater of 2200 µmol L⁻¹ to 2140 µmol L⁻¹ at the end of the experiment (Fig. 8h).

25



## 4 Discussion

At all stations, the incubations were carried out under typical oligotrophic ocean conditions, characterized by warm surface seawater temperatures (>20° C) low nutrient concentrations of $PO_4^{3-}$, $SiO_4^{4-}$, and $NO_3^-$, and low POC and low picophytoplankton cell abundances. Such conditions are typical for low-latitude oceans where strong thermal water

stratification leads to very low (re)supply of nutrients from the subsurface, resulting in N- and P-limited phytoplankton production (Moore et al., 2013). Therefore, the oligotrophic waters are dependent on nutrient input from the top via dust deposition or riverine input, or nutrient input from upwelling at the coast. Our observations from the original seawater showed an increase in $SiO_4^{4-}$ concentration from M1 in the east (0.71 µmol $L^{-1}$, Fig. 4) to M4 in the west (1.03 µmol $L^{-1}$, Fig. S3), suggesting a fluvial input from the Amazon river in the western Atlantic as the plume water is characterized by high dissolved

silicon concentrations (Shipe et al., 2006). Concentrations of DFe and $PO_4^{3-}$ in the original sampled seawater showed highest values at M1 (1 nmol $L^{-1}$ and 0.03 µmol $L^{-1}$, Fig. 3 & 4), potentially due to eastward shoaling of the ML enabling it to be more easily mixed by the wind, as well as due to offshore-upwelling associated to the Guinea Dome (Siedler et al., 1992), and to enhanced Saharan dust input in the east proximal to the Saharan desert (Bory and Newton, 2000; Fischer et al., 2016; Korte et al., 2017). Depending on the season and distance to the dust source, dry (winter) or wet (summer) dust deposition predominates

(Duce et al., 1991; Schulz et al., 2012) with dry deposition generally being dominant close to source regions (Dulac et al., 1989) and wet deposition becoming more prominent with increasing distance from the source (Bergametti et al., 1989).

### 4.1 Dry versus wet deposition

Substantial differences in released nutrients ($PO_4^{3-}$, $SiO_4^{4-}$, DFe) were observed between the dry- and wet-dust deposition treatments. Results from the dust incubations at M1 show that wet dust deposition resulted in an immediate increase in these

nutrient concentrations, while dry dust deposition left the tested waters unchanged (Fig. 3 and 4). Our observations show that the wet Saharan dust used in this experiment i.e. from a dune and lake as precursor dust source, has the potential to deliver $PO_4^{3-}$, $SiO_4^{4-}$ and DFe if leached in $H_2SO_4$. This result agrees with previous seeding experiments conducted in the oligotrophic Mediterranean Sea, which also showed that atmospherically processed (wet) dust that underwent condensation/evaporation cycles including exposure to $HNO_3$ and $H_2SO_4$, led to the increase of dissolved inorganic phosphorus and $NO_3^-$ concentrations

in the oligotrophic waters (Pulido-Villena et al., 2010; 2014; Ridame et al., 2014), while unprocessed (dry) dust left the waters unchanged (Ridame et al., 2014). In our experiments, the $NO_3^-$ concentrations remained unchanged in both the dry and wet dust deposition (Fig. 5). This result suggests that there is no nitrogen leached from the dust itself, but that dust carries nitrogen from the atmosphere and releases it to the ocean once deposited, which is in agreement with previous studies reporting similar observations (Louis et al., 2015). Therefore, Saharan dust is likely to be an important vehicle for transferring N-nutrients from

the atmosphere to the seawater affecting primary productivity (Ridame et al., 2014).

### 4.2 Variations due to the amount of dust

Whereas wet dust deposition resulted in the release of nutrients ($PO_4^{3-}$, $SiO_4^{4-}$, DFe) into the tested seawater, this process was highly dependent on the amount of dust added. A clear increase of $PO_4^{3-}$ concentration was only observed in the experiment when dust was added in high amounts (Fig. 3b and c), while $SiO_4^{4-}$ and DFe showed elevated concentrations

already after the addition of a low amount of dust (Fig. 4b and c). This observation implies that Saharan dust deposition to the Atlantic Ocean needs to be high in order to deliver bioavailable nutrients for promoting phytoplankton growth. Earlier dust-addition experiments in the tropical Atlantic Ocean showed that 0.5 and 2 mg $L^{-1}$ Saharan dust stimulated nitrogen fixation by relieving its P and Fe co-limitation (Mills et al., 2004). Similarly, addition of 1.34 mg $L^{-1}$ Saharan dust stimulated growth of the phytoplankton community in the Atlantic Ocean at 40°N (Blain et al., 2004). In the Mediterranean Sea, where Saharan

dust deposition can also be high, enrichment with 0.8 mg $L^{-1}$ Saharan dust was sufficient to stimulate phytoplankton growth (Ridame et al., 2014). Depending on the location along our transect, dust deposition varied in space and time and generally





declined with increasing distance from the source (Korte et al., 2017). Despite the lower amounts of dust deposited in the western tropical North Atlantic compared to the eastern part, the increased contribution of wet dust deposition, reaching up to 1000 mg Saharan dust per m$^2$ from a single rain shower (Stuut, personal communication) may compensate for this. Therefore, Saharan dust, in particular wet deposition thereof, has the potential to introduce sufficient amounts of nutrients ($PO_4^{3-}$, $SiO_4^{4-}$,

DFe) to relieve nutrient co-limitation in the tropical Atlantic Ocean and to promote nitrogen fixation by diazotrophic cyanobacteria and ultimately provide new N-nutrients to other phytoplankton (Falkowski et al., 1998; Capone, 2001). Indeed, Mahaffey et al. (2003) showed that atmospheric dust deposition might drive the temporal variability of nitrogen fixation in the North Atlantic Ocean, and Pabortsava et al. (2017) interpreted diazotrophic activity in the North Atlantic gyre being stimulated by enhanced Saharan dust deposition.

**4.3 Variations related to different types of dust**

Wet dust incubation experiments at all stations revealed higher $SiO_4^{4-}$ and DFe concentrations leached from the lake dust compared to the dune dust but reached similar $PO_4^{3-}$ concentrations when leached from both dust types (Fig. 3 & 4). This result implies that different types of dust (i.e. lake deposit vs. dune sand), transported and processed in the atmosphere can influence the ocean biogeochemistry in distinct ways and possibly impact only particular phytoplankton groups each requiring

the leached nutrients in different amounts. While, for example, organisms producing biogenic silica like diatoms (class Bacillariophyceae) require $SiO_4^{4-}$ and DFe to thrive, $SiO_4^{4-}$ is less important for calcifying coccolithophores (class Haptophyceae) or nitrogen fixing cyanobacteria (class Cyanophyceae), which groups depend more on $PO_4^{3-}$ (Boyd et al., 2010). In our experiment neither of the dust types resulted in an increase of the plankton cells derived by the leached nutrients due to the missing nitrogen source as described above. However, seeding experiments in the Mediterranean Sea with

atmospherically processed dust from southern Tunisia (Guieu et al., 2010; Bressac et al., 2012; Giovagnetti et al., 2013; Ridame et al., 2014; Desboeufs et al., 2014) have shown that this dust type introduced nitrogen and/or nitrogen and phosphorus when it was exposed to $HNO_3$/$H_2SO_4$ condensation/evaporation cycles, relieving nutrient (co-)limitation of phytoplankton activity. The concentration of the picoplankton Haptophyceae, Cyanophyceao, and Pelagophyceae doubled after a first dust addition, while the biomass of nano- and microplankton Haptophyceae mainly increased after a second dust addition (Giovagnetti et al.,

2013). Bonnet et al. (2005) and Blain et al. (2004) showed that Saharan dust from the Hoggar regions in southern Algeria is a potential source of bioavailable Fe that stimulates mainly the picophytoplankton (< 2 μm) in the Mediterranean Sea and diatoms in the northeast Atlantic Ocean. Observed under natural conditions, Guerreiro et al. (2017) reported a flux increase of opportunistic coccolithophore species during the fall 2013 in the western tropical North Atlantic in combination with a wet dust deposition event, suggesting that calcifying phytoplankton also benefited from Saharan dust-born nutrients in this region.

**4.4. Impact of pH on bioavailablity of dust-born nutrients'**

Nutrient release ($PO_4^{3-}$, $SiO_4^{4-}$, DFe) was strongly dependent on the pH, to the extent that nutrients were leached from the dust only in pH 2 rain (Fig. 3, 4, and 7). In contrast, when dust was leached in pH 4.5 rain, all nutrient concentrations remained as low as when dry dust was added, and the nutrient development showed a similar temporal progression as observed in the control samples (Fig. 8). Such pH dependency is in agreement with Spokes et al. (1994), who showed that highest metal

solubility found under low pH conditions. Accordingly, atmospheric processing under a low pH regime will increase the nutrient's solubility during transport (Ridame and Guieu, 2002; Mahowald et al., 2005; Baker and Croot, 2010), which is needed to introduce bioavailable nutrients to phytoplankton in the surface waters.

In our experiments, enhancement of nutrient concentration of $PO_4^{3-}$, $SiO_4^{4-}$ and DFe was observed immediately after wet dust addition in pH 2 rain at station M1 (Fig. 3 and 4), and at station M3 in the ML and DL for $SiO_4^{4-}$, and DFe (Fig. 7

and 8). During the experiments, the nutrient concentrations remained stable ($SiO_4^{4-}$) or decreased ($PO_4^{3-}$ and DFe), while there was no significant increase observed in picoplankton cell abundance (Fig.6, 7, and 8). Despite the initial increase of





*Synechoccus* spp. from the original seawater after the day at station M1 and M3, the temporal dynamic was followed by a subsequent decrease until the end of the experiments, except for the slight increase in the experiment with the DL at M3 (Fig. 8e) and the unidentified picoeukaryotic phytoplankton group (*R3*) in the ML at station M3 (Fig. 7f). This contrasting observation suggests an abiotic decrease of nutrient concentration rather than biological uptake. In the case for DFe, the higher
pH of the seawater (~ 8) leads to precipitation of iron (Spokes and Jickells, 1996), while the dissolved inorganic phosphate might be scavenged by the dust particles present in the incubation bottles (Louis et al., 2015).

The pH of the rain also affected the DIC concentrations in the water, which decreased more strongly with pH 2 rain and high amounts of dust (Fig. 6b and c) and less strongly with pH 4.5 rain (Fig. 8h). However, in all experiments, changes were too small for consider the change as being caused by a biological response.

**4.5 The role of dust for POC aggregate formation**

Scavenging of organic material by dust particles may also play a role in forming POC aggregates. Although the dry-dust deposition treatments did not lead to nutrient release in our experiments, the resulting POC concentrations were as high as after wet dust deposition (Fig. 3). Interestingly, despite the large standard variation, the average POC concentrations increased from the rain-control sample (0.05 mg POC $L^{-1}$), to low wet-dust deposition (0.06 mg POC $L^{-1}$) to high wet-dust
deposition (0.07 mg POC $L^{-1}$), and high dry-dust deposition (0.07 mg POC $L^{-1}$). This observation suggests that the suspended dust particles may become incorporated in aggregates formed by the organic matter already present in the original seawater. It seems that the amount and possibly also the size of the so-called marine snow aggregates will increase with higher amounts of dust added, via either wet or dry dust deposition. After termination by filtration, these newly formed aggregates were retained on the GF/F filter (pore size 0.7 µm) but single particles would have passed through without dust-stimulated aggregation.
Experiments dealing with the measurement of POC export fluxes showed that dust triggered abiotic formation of transparent exopolymeric particles, which eventually lead to the formation of sinking organic matter (Louis et al., 2017), resulting in enhanced POC fluxes that were highly correlated to the mineral dust flux (Bressac et al., 2014; Desboeufs et al., 2014; Louis et al., 2017). Also, dust addition experiments in roller tanks revealed that Saharan dust leads to the formation of marine snow aggregates within the first 12 hours after its addition (Van der Jagt et al., 2018). These experiments also showed that the size-
specific sinking velocities of these aggregates depended on the amount of added dust. The same authors argue that aggregate formation already takes place in the surface waters and no additional particles are scavenged when settling to greater depths (Van der Jagt et al., 2018). Given our experimental results, this would mean that there is no abiotic increase in POC concentrations in the deep layer caused by dust input. However, our experiment at M3, clearly showed the four-fold increase of POC concentrations in the lower ML at the end of the experiment (Fig. 8g). Therefore, the already high original nutrient
concentrations in the DL, which decreased during the experiment (Fig. 8a-d), may have biologically produced the high POC concentrations. Also, the increase of *Synechococcos* spp. in all treatments of the DL argue for picophytoplankton growth, albeit most likely not dust-induced since $PO_4^{3-}$ concentrations did not increase after dust addition (Fig. 8a), though required by cyanobacteria (Boyd et al., 2010). Nevertheless, even without a dust-induced fertilizing effect, dust particles suspended in the water column may have contributed to the formation of marine snow aggregates, contributing to enhance POC export fluxes
(Ternon et al., 2010; Bressac and Guieu, 2013). Unfortunately, export fluxes cannot be measured in bottle experiments and should therefore be studied in subsequent experiments.

**4.6 Further implications**

Our experiments show significant differences in nutrient release between wet and dry dust deposition with important implications for primary production and the biological carbon pump in the tropical North Atlantic Ocean, where massive
amounts of Saharan dust are deposited (Moulin et al., 1997; Prospero, 1999; Yu et al., 2015). We showed that wet deposition of Saharan dust will deliver $PO_4^{3-}$, $SiO_4^{4-}$ and DFe to the surface waters if processed at low pH (pH=2) and if dust deposition



fluxes are high ($> 1.5$ mg L$^{-1}$). Considering a scenario in which climate is changing, and anthropogenic emissions, global temperatures and possibly desertification are increasing, such combination of factors is likely to result in increasingly higher concentrations of anthropogenic $HNO_3$ and $H_2SO_4$ in the atmosphere in tandem with higher amounts of dust deflated from the continental deserts. Such high amounts of dust, which are processed in the atmosphere and deposited in the surface oceans,

will stimulate not only the growth of the autotrophic phytoplankton community but also that of heterotrophic bacteria (Herut et al., 2005; Guieu et al., 2014b), two processes having opposing effects in terms of atmospheric $CO_2$ regulation. In addition, dust contributes to the production of carbon-rich aggregates and acts as ballast material for promoting the sinking of (newly) formed organic matter (Bressac et al., 2012; Louis et al., 2017). Combining all these effects, Saharan dust is likely to play a key role in either enhancing or decreasing atmospheric $CO_2$, impacting heterotrophic or autotrophic activity, respectively. This

illustrates the importance of determining the relative contributions of each of these different processes and assessing the flux of organic matter being transported from the upper to the deep ocean without degradation. Consequently, seeding experiments in the North Atlantic Ocean quantifying relevant environmental parameters, i.e. an atmospheric nitrogen source, carbon and nitrogen uptake by the autotrophic and heterotrophic community, and especially particle export fluxes, are needed to address these issues.

**5 Conclusions**

        Wet and dry dust deposition incubation experiments were carried out along a transatlantic transect in the tropical North Atlantic Ocean. Results showed that both wet and dry dust deposition had an effect on the ocean biogeochemistry. High amounts of dust leached in artificial acidified rain ($H_2SO_4$, pH=2) led to higher nutrient concentrations released into the seawater opposed to dry deposited dust for which the nutrient concentrations remained unchanged. Our observations show that

only wet dust deposition is likely to release enough bioavailable nutrients (i.e. $PO_4^{3-}$, $SiO_4^{4-}$, DFe) to trigger phytoplankton growth. Clear evidences for phytoplankton growth in our experiments is lacking, most likely due to a missing source of new nitrogen-nutrients. Nonetheless, the deposited dust – dry and wet – resulted in equally high POC concentrations at the end of the experiment, pointing to an important role of dust in stimulating organic matter aggregation. Therefore, even without acting as a fertilizer, dust likely enhances the export of particulate organic matter from the upper ocean down to the deep North

Atlantic Ocean by promoting the formation of aggregates and acting as a ballast mineral, with potential implications for the marine biological carbon pump.

**Acknowledgements**

The project was funded by ERC (project no. 311152: DUSTTRAFFIC) and NWO (project no. 822.01.008: TRAFFIC). FP
and ST were funded by the Helmholtz association (HGF Young Investigators Group EcoTrace, VH-NG-901). The authors
would like to thank the captain and crew of *RSS James Cook* during cruise JC 134, the shipboard NIOZ technicians Barry
Boersen and Yvo Witte, and scientists Dirk Jong, Oliver Knebel, Monica Martens and Anne Roepert as well as the technical
and analytical support given on board and at NIOZ by Karel Bakker, Tessa de Bruin, Kirsten Kooijman, Patrick Laan, Anna
Noordeloos and Sharyn Ossebaar.




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
