# Peer review of "Effects of dry and wet Saharan dust deposition in the tropical North Atlantic Ocean"

_Biogeosciences, 2018_

## Referee Comment (RC1) · Anonymous Referee #1 · 11 Jan 2019

This manuscript presents results obtained during incubation experiments performed with seawater collected at different sites in the tropical North Atlantic Ocean, and submitted to different types (concentrations, wet vs. dry, dust source regions) of dust additions. The release of nutrients and the subsequent response of pico-phytoplankton were followed for 4-8 days. The initial design of this experiment is interesting and robust with different treatments (different dust concentrations and mode of deposition), systematic controls and three replicate incubation bottles per treatment. However, while this topic is timely, some important information are missing and I think that the obtained dataset does not allow the authors to tackle the main problematic of this study, i.e., the potential of Saharan dust as a fertilizer for phytoplankton growth. I think that this manuscript should be rewritten in order to focus on the main findings, i.e., the

absence of N release and the non-response of the pico-phytoplankton community.

Main comments - Three incubation experiments have been performed. Only two are discussed and the third one can be found in the supplementary info. Since results from the third experiment are not discussed at all, I would suggest to remove it from the manuscript, or at least from the abstract. - Cell abundance measured by flow cytometry is the only parameter used to follow the biological response. Information about chlorophyll a, micro-phytoplankton, etc. are missing. For example, the decrease in Si and increase of POC (Fig. 8) seem to indicate a response of the diatom community rather than the formation of aggregates. - The biological response is not mentioned in the discussion section while it represents the main problematic of this study. I understand that this is probably due to the lack of evidence of a fertilization effect. However, this experiment is neither designed to investigate/quantify the release of nutrients, nor the aggregation process. - I suggest to remove the section 4.5 about the aggregation process. Only final POC concentrations are used to discuss this process. How the authors discriminate (and quantify) newly formed aggregates from the increase in micro-phytoplankton cells for example? How did the incubation conditions influence the formation of aggregates? Were the particles maintained in suspension or did they sit on the bottom of the bottles?

Additional comments The title should be modified to be more precise. Abstract - L29-30 – not necessary to specify M1 and M3 as there are no additional information for these sites in the abstract. The increase in Synechococcus was probably not related to the additions of dust since the same increase was observed in the control treatments. Table 1 – I suggest to replace "mg" by "mg/L". P8-L24 – Replace "0 uM" by "below the detection limit". P11-L14-16-24 – I suggest to replace "original" by "initial". P14-L33 – I suggest to replace "nutrient development showed a similar temporal progression" by "nutrient development showed a similar temporal evolution". Figures 7 and 8 – Which incubation experiment? Finally, some parameters presented in this manuscript are not discussed, e.g., DIC.

---

## Referee Comment (RC2) · Anonymous Referee #2 · 21 Jan 2019

**General comments**

Mineral dust transported in the atmosphere from arid continental landmasses to the oceanic realm represents a potential supply of bio-limiting nutrients for marine ecosystems. Mineral dust is therefore thought to play a key role in the open ocean biological productivity, and could also enhance carbon export down through the water column due to its contribution to the ballasting of marine particulate matter. The impact of dust on primary production is expected to be particularly significant in HNLC areas where iron is the main limiting nutrient for phytoplankton growth. In oligotrophic regions where phytoplankton development is controlled by phosphorus and nitrogen availability such as the Tropical Atlantic ocean, mineral dust could also boost productivity by stimulating nitrogen fixation. Yet, the impact of Saharan dust inputs across the Tropical Atlantic

(by far the largest mineral dust delivery to the ocean) on surface waters productivity is insufficiently documented, and it is still unclear how significant the biogeochemical impact of Saharan dust is. In this manuscript, Korte et al. report on incubation experiments conducted along a trans-Atlantic transect at about 12°N and designed to further our understanding of the effect of dust delivery on nutrients release, phytoplankton response and particulate organic matter production. As earlier studies, in the Mediterranean in particular, have suggested that the deposition mode (wet vs dry) could have an influence on the nutrient release from the dust, the authors tested the biogeochemical impact of both dry and wet deposition of mineral dust on various Atlantic waters from 23° to 49°W sampled at various water depth. Different quantities of dust (submitted -or not- to acidified artificial rain mimicking atmospheric conditions), from two distinct West African sources, were added to seawater to determine whether these factors may influence the response of the ocean biogeochemistry to the dust delivery. Many parameters were analyzed including nutrients (PO3-, NO3-, SiO44-, dissolved iron), particulate organic carbon, and picoplankton abundances. Incubation experimental studies are tricky to set up and so such an extensive effort must therefore be commended. This experiment therefore yielded some important advance for our understanding of the potential impact of Saharan dust on the Atlantic surface water biological productivity. Among other findings, this study confirms the fundamental role of the atmospheric pre-conditioning of the dust (through acidic cocktails) to allow for nutrients release (PO3-, SiO44-, dissolved iron) and potential impact on ocean biogeochemistry; also, this study highlights the importance of the dust atmospheric cycle (and its contact with HNO3) for nitrogen release (in these incubation experiments, dust inputs did not result in nitrogen release as the dust introduced in the incubation bottles had not been subjected to atmospheric pre-conditioning). Furthermore, according to the authors, the amount of wet-deposited dust to the Atlantic might be sufficient for biological stimulation via nutrient release, even in the western part of the ocean where dust inputs is much lower than on the eastern side of the basin. Another interesting outcome is that the two different types of dust used in the experiments (from two different dust sources
in West Africa) yielded distinct SiO44- and dissolved iron (while releasing similar phosphate amounts), suggesting dust from different sources may have dissimilar impacts on the ocean biogeochemistry. These outcome should be particularly useful for the set up of seeding experiments in the Atlantic ocean. Also, I find the manuscript well organized, clearly written and appropriately illustrated. I would therefore recommend publication in BG nearly as is. Still, I have listed a few comments/questions below that I hope will be of some use to the authors while working on the final version of the manuscript.

Specific comments

page 3, line 20: why would clay material be expected to contain more bioavailable nutrients than coarser (supposedly less weathered?) material?

pages 4-5, bridging sentence : it is unclear to me what is the reasoning for the addition of 40mL of artificial rainwater (in about 6 liters?) and how this translates into a precipitation rate of 0,04 mm d-1; could you please clarify?

page 13, line 9: is there any evidence (other than the SiO44- concentration) of an Amazonian influence all the way to station M3 in the middle of the Atlantic?

page 13, lines 33 and 34 : I find interesting that the increase of PO43- is only observed when dust was added in large amounts; wouldn't the relationship between dust and PO43- be expected to be linear (assuming the dust samples were well homogenized)? what could possibly explain the existence of such an apparent threshold for PO43-release?

page 14, line 6 : if the release of PO43- and dissolved iron may promote nitrogen fixation by diazotrophic cyanobacteria, why there was no such response by diazotrophic species in the incubation bottles?

page 15, line 4: the "abiotic" hypothesis for the decrease in nutrient concentration through the experiment raises the question of the bioavailability of the released nutri-

BGD
ents; if, as indicated in the text (quoting earlier studies), the elevated pH of seawater leads to iron precipitation for instance, is the precipitation kinetic known and will iron be available long enough to be used by the phytoplankton?

page 15, lines 33-34: the fact that there is no difference between the incubation bottles and the control bottle does not favor a major role of the dust in the formation of marine snow aggregate, does it?

page 16, line 23-24: again, this seems to me a bit of an overstatement as a significant POC increase in only observed at station M3, and that, in all cases, the incubation bottles do not show significant differences from the control bottles

**Technical corrections**

page 5, table 1: shouldn't dust addition unit be mg.L-1

page 6, section 3.1.1: there seems to be a bit of redundancy between the first and second paragraphs

---

## Author Comment (AC2) · 11 Feb 2019

**General comments**

Mineral dust transported in the atmosphere from arid continental landmasses to the oceanic realm represents a potential supply of bio-limiting nutrients for marine ecosystems. Mineral dust is therefore thought to play a key role in the open ocean biological productivity, and could also enhance carbon export down through the water column due to its contribution to the ballasting of marine particulate matter. The impact of dust on primary production is expected to be particularly significant in HNLC areas where iron is the main limiting nutrient for phytoplankton growth. In oligotrophic regions where phytoplankton development is controlled by phosphorus and nitrogen availability such as the Tropical Atlantic ocean, mineral dust could also boost productivity by stimulating nitrogen fixation. Yet, the impact of Saharan dust inputs across the Tropical Atlantic (by far the largest mineral dust delivery to the ocean) on surface waters productivity is insufficiently documented, and it is still unclear how significant the biogeochemical impact of Saharan dust is. In this manuscript, Korte et al. report on incubation experiments conducted along a trans-Atlantic transect at about 12°N and designed to further our understanding of the effect of dust delivery on nutrients release, phytoplankton response and particulate organic matter production. As earlier studies, in the Mediterranean in particular, have suggested that the deposition mode (wet vs dry) could have an influence on the nutrient release from the dust, the authors tested the biogeochemical impact of both dry and wet deposition of mineral dust on various Atlantic waters from 23°to 49°W sampled at various water depth. Different quantities of dust (sub-mitted -or not- to acidified artificial rain mimicking atmospheric conditions), from two distinct West African sources, were added to seawater to determine whether these factors may influence the response of the ocean biogeochemistry to the dust delivery. Many parameters were analyzed including nutrients ($PO_3^-$, $NO_3^-$, $SiO_4^{4-}$, dissolved iron), particulate organic carbon, and picoplankton abundances. Incubation experimental studies are tricky to set up and so such an extensive effort must therefore be commended. This experiment therefore yielded some important advance for our under-standing of the potential impact of Saharan dust on the Atlantic surface water biological productivity. Among other findings, this study confirms the fundamental role of the atmospheric pre-conditioning of the dust (through acidic cocktails) to allow for nutrients release ($PO_3^-$, $SiO_4^{4-}$, dissolved iron) and potential impact on ocean biogeochemistry; also, this study highlights the importance of the dust atmospheric

cycle (and its contact with HNO3) for nitrogen release (in these incubation experiments, dust inputs did not result in nitrogen release as the dust introduced in the incubation bottles had not been subjected to atmospheric pre-conditioning). Furthermore, according to the authors, the amount of wet-deposited dust to the Atlantic might be sufficient for biological stimulation via nutrient release, even in the western part of the ocean where dust inputs is much lower than on the eastern side of the basin. Another interesting outcome is that the two different types of dust used in the experiments (from two different dust sources) yielded distinct $SiO_4^{4-}$ and dissolved iron (while releasing similar phosphate amounts), suggesting dust from different sources may have dissimilar impacts on the ocean biogeochemistry. These outcome should be particularly useful for the set up of seeding experiments in the Atlantic ocean. Also, I find the manuscript well organized, clearly written and appropriately illustrated. I would therefore recommend publication in BG nearly as is. Still, I have listed a few comments/questions below that I hope will be of some use to the authors while working on the final version of the manuscript.

Thank you for the overview into the fertilization topic and your overall positive review and validation of our incubation experiment. As Reviewer #1 had comments on the aggregate formation and the first experiment we conducted in the western Atlantic with the suggestion to remove these results, we decided to do so and shortened the manuscript to gain more focus on the nutrient release findings.

But of course, we are happy to reply to your specific and technical comments as stated below.

**Specific comments**

page 3, line 20: why would clay material be expected to contain more bioavailable nutrients than coarser (supposedly less weathered?) material?

We clarified our line of thinking by adding an extra sentence on the mineralogy of the two dust types.

'The lake dust was expected to contain more bioavailable nutrients, e.g. silicate and dissolved iron, since lake deposits are often associated with freshwater diatoms and fine-grained iron-containing clay sediments (Scheuvens et al., 2013; Bristow et al., 2010). The dune dust consisted of coarser-grained sediments associated with more refractory minerals like quartz and feldspar.'

pages 4-5, bridging sentence: it is unclear to me what is the reasoning for the addition of 40mL of artificial rainwater (in about 6 liters?) and how this translates into a precipitation rate of 0,04 mm d-1; could you please clarify?

Yes, we agree that the calculation is unclear.

We used the satellite-derived precipitation data at the specific locations of the transect. While it rains most in the western Atlantic and least in the eastern Atlantic, we chose the average amount of rain in the centre of the transect (M3, 0.05 mm d$^{-1}$). Over 1m$^2$ of ocean, this amount would translate to 0.05 L (1m*1m*0.00005m=0.00005m$^3$=0.05L). As the eastern Atlantic receives less rain, we decided to stay below this amount to not overestimate the precipitation rate.

We added this information in the paragraph 2.3 as follows:

'The amount of rainwater was chosen based on satellite-derived precipitation data. During spring, the time of the year in which the incubation experiments were conducted, it rains most in the western Atlantic, while it rains least in the eastern Atlantic (Fig. S2). Given the average precipitation rate of 0.049 mm per day at M3 in the centre of the transect, it would translate to 50 mL of rain per 1 m$^2$ of water. Therefore, we stayed below this amount to not overestimate the rain. According to Van der Does et al. (2016), a small amount of precipitation is already sufficient to wash out suspended dust from the atmosphere by wet deposition.'

page 13, line 9: is there any evidence (other than the SiO44- concentration) of an Amazonian influence all the way to station M3 in the middle of the Atlantic?

Next to high silicate concentrations, the Amazon River affected water is also low in salinity. During the times when the Amazon River discharge is retroflected into the North Equatorial Counter Current between June and January, the surface water salinities decrease eastward in the open ocean. With salinity observations, it would be possible to trace the Amazonian influence all the way to station M4 in the west, and possibly even to station M3 in the central Atlantic. However, during the time of the incubation experiment in March, the Amazon River discharge was transported with the North Brazil Current in north-western direction along the coast of Brazil. Therefore, we do not mention the possibility of lower salinities in the surface waters by the Amazon River in the manuscript.

page 13, lines 33 and 34: I find interesting that the increase of PO43- is only observed when dust was added in large amounts; wouldn't the relationship between dust andPO43- be expected to be linear (assuming the dust samples were well homogenized)? what could possibly explain the existence of such an apparent threshold for PO43-release?

The phosphorous concentrations do slightly increase with the low dust amounts as well, however the increase is insignificant from the control sample. When looking at the nutrient release after wet dust addition for phosphate, silicate and dissolved iron, there is a linear relationship for both dust types (Fig. below).
We added this figure in the supplement (Fig. S3) and added the information in the paragraph 4.2 in the discussions as follows:

'Although the nutrient release of dust was linear to the dust amounts added (Fig. S3), a significant increase of $PO_4^{3-}$ concentration was only observed in the experiment when dust was added in high amounts (Fig. 3b and c), while $SiO_4^{4-}$ and DFe showed significant elevated concentrations already after the addition of a low amount of dust (Fig. 4b and c).

[Figure]

page 14, line 6: if the release of $PO_4^{3-}$ and dissolved iron may promote nitrogen fixation by diazotrophic cyanobacteria, why there was no such response by diazotrophic species in the incubation bottles?

page 15, line 4: the "abiotic" hypothesis for the decrease in nutrient concentration through the experiment raises the question of the bioavailability of the released nutrients; if, as indicated in the text (quoting earlier studies), the elevated pH of

seawater leads to iron precipitation for instance, is the precipitation kinetic known and will iron be available long enough to be used by the phytoplankton?

We could like to reply to the comments on the diazotrophic response and iron kinetic together since they are coupled.

It might be that we do not see a response by diazotrophic species since they demand a high iron concentration in the seawater to thrive. Although the iron concentrations increase extremely in the beginning of the experiment with wet dust deposition, they also decrease throughout the experiment due to complex binding ligands and precipitation in seawater pH (~8). There are also scavenging processes of phosphate back to the dust particles decreasing its bioavailability. As a detailed assessment of the iron kinetics is beyond our expertise as well as the scope of this paper, we can only speculate why the increase in dissolved iron did not lead to an increase in diazotrophic respond. We merely want to demonstrate that several metals and nutrients are leached off dust particles during droplet formation, thus potentially increasing their bioavailability with wet deposition.

We added the following sentences in paragraph 4.4 in the discussions.

'This contrasting observation of nutrient decrease without cell abundance increase, suggests an abiotic decrease of nutrient concentration rather than biological uptake. Diazotrophs are in need of high iron concentrations to thrive (Berman-Frank et al., 2001). Although the DFe concentrations were extremely high at the beginning of the experiment, the iron will be complexed by organic ligands in the seawater (Rue and Bruland, 1995) and the higher pH of the water (~ 8) leads to iron precipitation (Spokes and Jickells, 1996), reducing its bioavailability. In addition, the observed decrease of dissolved inorganic phosphate might be due to scavenging processes back to the (iron-containing) dust particles in the incubation bottles (Louis et al., 2015), inhibiting biological uptake.'

And added the information of abiotic processes in the conclusions.

'Clear evidences for phytoplankton growth in our experiments is however lacking, possibly due to a missing source of new nitrogen-nutrients and abiotic nutrient precipitation processes.'

page 15, lines 33-34: the fact that there is no difference between the incubation bottles and the control bottle does not favor a major role of the dust in the formation of marine snow aggregate, does it?

To determine the role of dust in aggregation formation in our incubation experiments is only speculative. Within our incubation experiment it was not possible to look at the aggregates individually, e.g. if they contained dust. We only judged from the final POC concentrations at M1 that dust might be responsible for aggregate formation since final POC concentrations increase with added dust amounts. However, at M3 this simple relationship is not valid anymore as there is, as you noticed, no difference between the control and dust addition treatment. As

Reviewer #1 also mentions that we cannot discriminate and quantify between newly formed aggregates and increase in micro-phytoplankton cells, we decided to remove the entire paragraph 4.5 as a more detailed analysis on aggregate formation is needed to identify the role of dust.

page 16, line 23-24: again, this seems to me a bit of an overstatement as a significant POC increase in only observed at station M3, and that, in all cases, the incubation bottles do not show significant differences from the control bottles

We agree and we removed the speculations on POC formation as mentioned in the comment above.

**Technical corrections**

page 5, table 1: shouldn't dust addition unit be mg L-1

Yes, that is true and fixed.

page 6, section 3.1.1: there seems to be a bit of redundancy between the first and second paragraphs

The paragraph might read a bit monotonous, but we think that the comparison between nutrient release of the lake and dune dust gets easier to follow this way.

---

## Author Comment (AC1)

This manuscript presents results obtained during incubation experiments performed with seawater collected at different sites in the tropical North Atlantic Ocean, and submitted to different types (concentrations, wet vs. dry, dust source regions) of dust additions. The release of nutrients and the subsequent response of pico-phytoplankton were followed for 4-8 days. The initial design of this experiment is interesting and robust with different treatments (different dust concentrations and mode of deposition), systematic controls and three replicate incubation bottles per treatment. However, while this topic is timely, some important information are missing and I think that the obtained dataset does not allow the authors to tackle the main problematic of this study, i.e., the potential of Saharan dust as a fertilizer for phytoplankton growth. I think that this manuscript should be rewritten in order to focus on the main findings, i.e., the absence of N release and the non-response of the pico-phytoplankton community.

Thank you for taking your time to review and comment on our manuscript about the nutrient release of Saharan dust in the Atlantic Ocean.

Our experiment was designed to determine biogeochemical effects of dry and wet Saharan dust deposition in the oligotrophic waters of the Atlantic Ocean without adding nitrogen from another source other than Saharan dust. Therefore, we omitted e.g. nitric acid as a N-nutrient source in the wet deposition treatments to assess the potential response of the pico-phytoplankton community from only nutrients of Saharan dust. Such a respond was not observed. However, we found contrasting nutrient releases of phosphate, silicate and dissolved iron of the two dust types we have used, which is what we concentrated our manuscript on.

Still, we are very grateful for your comments and suggestion that we adopted as listed below.

Main comments

Three incubation experiments have been performed. Only two are discussed and the third one can be found in the supplementary info. Since results from the third experiment are not discussed at all, I would suggest to remove it from the manuscript, or at least from the abstract.

Yes, we agree with the comment given by the reviewer. It is right that we conducted three incubation experiments from which we could only discuss two experiments since the first from the western Atlantic was not successful. Still, we considered it beneficial to the scientific community to show the results from all the experiments. As the data are now already published in the discussion paper, and to increase the readability of the paper, we decided to remove the results from the manuscript.

Cell abundance measured by flow cytometry is the only parameter used to follow the biological response. Information about chlorophyll a, micro-phytoplankton, etc. are missing. For example, the decrease in Si and increase of POC (Fig. 8) seem to indicate a response of the diatom community rather than the formation of aggregates. The biological response is not mentioned in the discussion section while it represents the main problematic of this study. I understand that this is probably due to the lack of evidence of a fertilization effect. However, this experiment is neither designed to investigate/quantify the release of nutrients, nor the aggregation process.

Indeed, the cell abundances, POC and nutrients are the only parameters we followed to constrain biological responses. During our experiments we tried to filter water for biogenic silica analysis, which was unfortunately unsuccessful. The data we have for silicate is therefore the dissolved nutrient measurements, which indeed decrease in the middle of the Atlantic.

To address the reviewer's point, we added one sentence in paragraph 4.3 of the discussions on the possible response of the diatom community.

'In our experiments in the DL at M3 (Fig. 8), there was a nutrient decrease of $SiO_4^{4-}$ (Fig. 8b), in tandem with a POC increase (Fig. 8g), suggesting uptake by the diatom community, although the dust addition did not result in an obvious increase of the plankton cells (Fig. 8e, f).'

I suggest to remove the section 4.5 about the aggregation process. Only final POC concentrations are used to discuss this process. How the authors discriminate (and quantify) newly formed aggregates from the increase in micro-phytoplankton cells for example? How did the incubation conditions influence the formation of aggregates? Were the particles maintained in suspension or did they sit on the bottom of the bottles?

According to the reviewer's suggestion, we removed the paragraph about the aggregation process. The incubated 6 litres of water allowed us to only analyse the final concentrations of POC. Since these concentrations increased with increasing dust amounts added, we speculated about aggregation processes. In the bottles the

aggregates were in suspension, or at least brought into suspension once a day for nutrient and cell count sampling. However, we agree that we cannot make a solid argument about dust aggregation processes and have therefore removed the section from the manuscript.

Additional comments

The title should be modified to be more precise.

We changed the title to

'Nutrient release from dry and wet Saharan dust deposition in the tropical North Atlantic Ocean'

Abstract - L29-30 – not necessary to specify M1 and M3 as there are no additional information for these sites in the abstract.

We removed the station names in the abstract.

'After an initial increase in cell abundance, a subsequent decrease of these was observed for all experiments, independently of dry- and wet-dust deposition.'

The increase in Synechococcus was probably not related to the additions of dust since the same increase was observed in the control treatments.

Yes, we made it more precise in the same sentence as above.

'After an initial increase in cell abundance, a subsequent decrease of these was observed for all experiments, independently of dry- and wet-dust deposition.'

Table 1 – I suggest to replace "mg" by "mg/L"

We made the suggested change in the revised manuscript.

P8-L24 – Replace "0 uM" by "below the detection limit".

As suggested, we replaced 0 µmol L$^{-1}$ with below the detection limit.

'Concentrations in all treatments were below the detection limit and up to 1 µmol L$^{-1}$ with large error bars pointing to inhomogeneity in the three replicates.'

'During the 8 days of the experiment, the concentrations were also below the detection limit and up to 1.2 µmol L$^{-1}$ with irregular peaks and large error bars (Fig. 5c).'

P11-L14-16-24 – I suggest to replace "original" by "initial".

We used 'original' for all the CTD baseline. Therefore, we like to be consistent and keep it like this throughout the manuscript. Otherwise it might get confusing with the 'initial' nutrient concentrations measured at M1 directly after dust addition at day 0.

P14-L33 –I suggest to replace "nutrient development showed a similar temporal progression" by"nutrient development showed a similar temporal evolution".

We replaced the phrase as supposed.

'In contrast, when dust was leached in pH 4.5 rain, all nutrient concentrations remained as low as when dry dust was added, and the nutrient development showed a similar temporal evolution as observed in the control samples (Fig. 8).'

Figures 7 and 8 – Which incubation experiment?

Figures 7 and 8 are the incubation experiments at M3. It is now stated in the figure captions.

Finally, some parameters presented in this manuscript are not discussed, e.g., DIC

Since the DIC values do not show any significant results, but were still analysed, we initially showed them in the manuscript. However, we agree with the reviewer that they are not discussed and therefore, we removed them from the manuscript.